# Disease Resistant Bouquet Vine Varieties: Assessment of the Phenolic, Aromatic, and Sensory Potential of Their Wines

**DOI:** 10.3390/biom9120793

**Published:** 2019-11-27

**Authors:** M. Reyes González-Centeno, Kleopatra Chira, Clément Miramont, Jean-Louis Escudier, Alain Samson, Jean-Michel Salmon, Hernan Ojeda, Pierre-Louis Teissedre

**Affiliations:** 1Univ. Bordeaux, ISVV, EA 4577, Œnologie, 210 Chemin de Leysotte, 33140 Villenave d’Ornon, France; reyes.gonzalez@u-bordeaux.fr (M.R.G.-C.); kleopatra.chira@u-bordeaux.fr (K.C.); clement.miramont@u-bordeaux.fr (C.M.); 2INRA, ISVV, USC 1366 Œnologie, 210 Chemin de Leysotte, 33140 Villenave d’Ornon, France; 3Unité Expérimentale de Pech Rouge (UE 0999), INRA, Domaine de Pech Rouge, 11430 Gruissan, France; jean-louis.escudier@inra.fr (J.-L.E.); alain.samson@inra.fr (A.S.); jean-michel.salmon@inra.fr (J.-M.S.); hernan.ojeda@inra.fr (H.O.)

**Keywords:** bouquet vines varieties, disease resistance, hybrid grapes, wine, phenolic composition, fruity aroma profile, sensory analysis

## Abstract

The search for grape varieties resistant to diseases and to climatic changes notably concerns the wine industry. Nine monovarietal wines from new red grape varieties resistant to cryptogamic diseases (downy and powdery mildews) were evaluated in terms of their total phenolic, anthocyanin and proanthocyanidin contents, anthocyanin profile, volatile composition, and sensory attributes. Thus, the question remains, will these hybrid grapes (≥97.5% of *Vitis vinifera* genome) lead to wines with organoleptic properties similar to those of *Vitis vinifera* wines that consumers are used to? Total phenolic (1547–3418 mg GA/L), anthocyanin (186–561 mg malvidin/L), and proanthocyanidin (1.4–4.5 g tannins/L) contents were in broad agreement with those previously described in the literature for monovarietal wines produced with well-known red grape varieties (Cabernet Sauvignon, Merlot, Syrah). With regard to fruity aroma, ethyl esters of straight-chain fatty acids (530–929 μg/L) stood out clearly as the major volatile components for all hybrid wines considered. Sensory analysis revealed significant differences (*p* < 0.05) for visual aspect, aroma, flavor, global balance, astringency, and body. Overall, these new hybrid grape varieties are not only resistant to cryptogamic diseases, but also present enough potential to become quality wines, since their phenolic and volatile attributes are close to those of common red monovarietal wines.

## 1. Introduction

Nowadays, certain winegrowing regions lack the climatic conditions to grow and produce traditional grape varieties consistently. Climate change significantly affects the quality, typicity, and production yield of grapes and wine. The main consequence on vineyards is the reduction of the vine vegetative cycle (earlier ripening date and precocity of grape harvest), which derives in a lower acidity and greater sweetness of grape berries, a stronger alcohol level, and reduced anthocyanin content of wines, as well as a modification of their aroma profile [1,2]. Furthermore, the susceptibility of traditional *Vitis vinifera* varieties to fungal diseases, such as downy and powdery mildews, has notably risen and the spraying of winegrowing areas has become a common practice less and less appreciated. However, there is an increasing social demand for sustainable development and reduced pesticide use [3]. The wine industry claims solutions to adapt to climate change, to control vine decay, to face weakness of some traditional cultivars, to avoid pesticide resistance, and to decrease grapevine susceptibility to cryptogamic diseases. For these reasons, current trends in viticulture and oenology focus on vine cross-breeding and the exploration of new hybrid grape varieties.

In an attempt to deal with *Vitis vinifera* susceptibility to cryptogamic diseases, crossing with certain American and Asian species has been conducted to transfer their resistance genes into the *Vitis vinifera* genetic background [4]. The idea was to create new hybrid grape varieties combining durable resistance to downy and powdery mildews with a berry quality suitable for the production of high quality wines [5]. In this context, in the 1970s, Alain Bouquet (INRA Montpellier) built a selected collection of new hybrid grape (HG) varieties resistant to the main cryptogamic diseases, both downy and powdery mildews. This pioneering work was conducted via four or five generations of backcrossing between *Muscadinia rotundifolia* (*Vitis rotundifolia*) and different *Vitis vinifera* grapevine varieties. All of the resulting grape varieties, so-called Bouquet varieties, presented ≥95% of the *Vitis vinifera* genome, a high level of resistance against downy and powdery mildews (presence of the resistance genes *RUN1* and *RPV1*), and good agronomic parameters (yield, growth) [6,7].

It is well known that the grape variety used to produce a particular wine plays an important role in its aroma, flavor, astringency, and color, due to the persistence of certain phenolic and aromatic markers, initially present in grape berries, throughout the entire process of winemaking [8]. Each grape variety presents its own phenolic fingerprint, as well as a particular antioxidant capacity and aroma precursors that could also significantly differ from one to another [9,10]. Thus, even if cryptogamic resistance has been demonstrated for certain hybrid grape varieties, especially with Bouquet references for 20 years now in the South of France (INRA Pech-Rouge and INRA Vassal), their oenological aptitude, compositional potential, organoleptic nature, and consumers’ acceptance should still be addressed. Research on the phenolic and volatile profile of wines produced from resistant hybrids may represent a significant step for supporting their promotion in winemaking.

The present research aims to elucidate the hidden phenolic, volatile, and sensory potentials of monovarietal red wines produced from new hybrid grape varieties. Then, the question remains as these wines exhibit typical organoleptic characteristics when compared to *Vitis vinifera* wines that consumers are used to.

## 2. Materials and Methods

### 2.1. Crossing and Hybrid Grape Production

Nine red Bouquet varieties among thirty were considered in the present research: HG-A, HG-B, HG-C, HG-D, and HG-E with 98.7% of *Vitis vinifera* genome; and HG-F, HG-G, HG-H, and HG-I with 99.2% of *Vitis vinifera* genome. Information about their backcross number and the grape varieties participating in their last backcross are specified in Table 1. Indeed, twelve among thirty are actually in inscription experimentation for 2023 with CIVL partner (Comité Interprofessionnel des Vins du Languedoc). 

All of them shared the same vineyard location, cultivation system, climate, soil type, vine cultivation practices since 2009, and harvesting time at the experimental unit of Pech Rouge from INRA (Gruissan, France).

### 2.2. Red Wine Vinification

Microvinification assays were carried out separately for each hybrid grape to obtain the corresponding monovarietal wines. Hybrid grapes were manually harvested at maturity during the 2016 vintage. Grapes were crushed and destemmed the day of harvest. Potassium metabisulphite (3 g/hL) was added during the transfer of must to a stainless steel tank (50 L) and *Saccharomyces cerevisiae* (Anchor NT 202, La Littorale) was included to perform alcoholic fermentation at 23–25 °C.

Wine was inoculated with lactic acid bacteria (Maloferm Fruity, Oenobrands) to perform malolactic fermentation (MLF). It was conducted in all cases at a constant temperature of 22 °C and extended for a variable period of time depending on the hybrid grape considered (from 7 to 35 d).

Once the MLF concluded (malic acid content ≤0.2 g/L), all wines were immediately racked, stabilized, and sulfitated prior to bottling and storage at 16 °C until further analysis.

Infrared spectrometry with Fourier transformation (FT-IR) was used to measure all of the classic oenological parameters of these wines in triplicate with a WineScanTM Flex (FOSS Analytical, Hillerød, Denmark).

### 2.3. Chromatic Parameters in Wines

Chromatic parameters of wines, i.e., absorbances at 420 (d420), 520 (d520) and 620 nm (d620) were spectrophotometrically determined in triplicate under 1 mm optical way with V-630 UV-Vis equipment (JASCO, Japan). The color intensity (CI, sum of the three absorbances), the hue (d420/d520) and the components yellow (d420%), red (d520%), and blue (d620%) were calculated.

### 2.4. Total Phenolics, Proanthocyanidins, and Anthocyanins Analyses

A modified Folin Ciocalteu method to be applied in 96-well microplates [11] was used to quantify total phenolic content of wines. All wines were diluted in water at a ratio 1:20 and an automated microplate reader (FLUOstar Optima, BMG LabTech, Champigny-sur-Marne, France) was used for the measurement. Results, expressed as mg of gallic acid equivalents per liter of wine, were a mean of six determinations.

Proanthocyanidin and anthocyanin contents of wines were also spectrophotometrically determined in triplicate, through the Bate–Smith reaction [12] and the sodium bisulfite discoloration method [13], respectively. For total proanthocyanidin measurement, wine diluted solutions were prepared at a ratio 1:50 with distilled water.

### 2.5. HPLC Analysis of Monomeric and Oligomeric Flavan-3-Ols

Wines, without any treatment, were filtered and injected directly in triplicate. Monomeric and oligomeric flavan-3-ol analysis was performed on a Thermo-Finnigan Surveyor HPLC system (Thermo Electron SAS, Villebon-sur-Yvette, France) consisting of an UV-Vis detector (Surveyor PDA Plus), a fluorescence detector (Surveyor FL Plus Detector), an autosampler (Surveyor autosampler Plus), and a quaternary pump (Surveyor MS pump Plus). The mobile phases were 0.5% (*v*/*v*) aqueous formic acid (solvent A) and 0.5% (*v*/*v*) formic acid in acetonitrile (solvent B). Separation was performed on a reversed-phase LiChrospher 100 RP18 (250 mm × 4 mm, 5 μm) column, by using the following binary elution system: 3% B at initial time for 3 min, a linear gradient from 3% to 5% B in 11 min, from 5% to 10% B in 8 min, from 10% to 14% B in 4 min, from 14% to 25% B in 14 min, from 25% to 100% B in 1 min, 100% B for 7 min to wash the column, a linear gradient from 100% to 3% B in 2 min, and then 3% B for 5 min to reequilibrate the system before the next injection. Flow rate was set at 1 mL/min, UV-Vis detection wavelength at 280 nm, and fluorescence detection at 280 and 320 nm, respectively, for excitation and emission wavelengths. (+)-catechin was used as external standard for calibration curves. Results were expressed as milligrams of catechin per liter of wine.

### 2.6. HPLC Analysis of Anthocyanins

Anthocyanin separation was performed according to the elution conditions, flow rate and composition of the mobile phases previously described by González-Centeno et al. [14]. Chromatographic analyses were carried out on an Agilent Nucleosil 100-5C18 (250 mm × 4.0 mm, 5 μm) column by using a Thermo-Accela HPLC instrument including a UV−Vis detector (Accela PDA detector), an autosampler (Accela autosampler), and a quaternary pump (Accela 600 pump). Wines were filtered and injected directly, with no prior treatment and/or dilution.

Anthocyanin 3-*O*-monoglucosides (delphinidin, Dp; cyanidin, Cy; petunidin, Pt; peonidin, Pn; and malvidin, Mlv), as well as the acetylated and *p*-coumaroylated forms of Pn and Mlv, were identified by comparison to injected external standards and/or previous results. All wines were analyzed in triplicate and results were expressed in mg of Mlv-3-*O*-monoglucoside per liter of wine.

### 2.7. Evaluation of the Total Antioxidant Capacity

To achieve a more reliable and complete outline of the antioxidant capacity of the wines, three different spectrophotometric assays were applied: ABTS, FRAP (Ferric Reducing Antioxidant Power), and CUPRAC (CUPric Reducing Antioxidant Capacity). Modified versions of the original antioxidant capacity assays were performed to fit these spectrophotometric analyses in 96-well microplates according to the procedures described by González-Centeno et al. [9].

In all cases, absorbance was determined at 25 °C with the same automated microplate reader used for total phenolic analysis, and Trolox (0–1.3 mM) was used as standard for the calibration curves. Wine diluted solutions were prepared at a ratio 1:20 with distilled water. Antioxidant capacity results were expressed as a mean of six determinations in mmols of Trolox equivalents per liter of wine (mM Trolox).

### 2.8. Volatile Composition of Wines: Extraction and Gas Chromatography Analysis

Esters, as main contributors to the fruity aroma profile of red wines, were quantified by adapting the gas chromatography methodology described by Antalick et al. [15]. Volatile extraction procedure performed prior to gas-chromatographic analyses, equipment and calibration conditions were all applied as previously specified by González-Centeno et al. [11]. Target compounds were identified by comparing their retention times and mass spectra with those of the pure reference standards. Selected ions were *m*/*z* 116 for ethyl isobutyrate; *m*/*z* 102 for ethyl propanoate and ethyl 2-methylbutyrate; *m*/*z* 88 for ethyl butanoate, ethyl hexanoate, ethyl octanoate, ethyl decanoate, ethyl dodecanoate, and ethyl 3-methylbutyrate; *m*/*z* 70 for isoamyl acetate; *m*/*z* 61 for propyl acetate; and *m*/*z* 56 for isobutyl acetate and butyl acetate. All samples were analyzed in triplicate. Calibration curves were established using pure reference standards analyzed under the same conditions than wine samples.

### 2.9. Sensory Analysis

Sensory analysis was performed by a panel of 15 expert enologists (9 males and 6 females). All evaluations were conducted in a standard sensory-analysis chamber [16], equipped with individual tasting booths, where a uniform temperature (19–22 °C) and source of lighting, absence of noise and distracting stimuli were guaranteed. Wines (30 mL) were presented in standard clear wine glasses [17], covered with a Petri dish to minimize the escape of volatile components and randomly coded with three-digit numbers.

Descriptive sensory analysis was conducted to assess the organoleptic profile of all wines. Judges first evaluated the orthonasal global balance and the fruitiness intensity, and then, after a short break, both taste (sweetness, bitterness, roundness) and tactile sensation (astringency). The selected panel was asked to rate the intensity level of all descriptors on a 6-point scale ranging from ‘absence’ (note 0) to ‘maximum intensity’ (note 6). Results of each descriptor were then expressed as the mean value of all the judges.

### 2.10. Statistical Analysis

All experimental results were reported as mean values with their corresponding standard deviations. Statistical analysis was conducted by the statistical package R version 3.5.1 (R Foundation for Statistical Computing, Wien, Austria). Normality and homocedasticity of the residuals were evaluated for all parameters, by using the Shapiro-Wilk test and Levene’s test, respectively. When populations were distributed normally and showed homogeneity in variance, the parametric ANOVA and Tukey tests were used to evaluate the existence and degree of significant differences. These statistical analyses were replaced, respectively, by the non-parametric Kruskal–Wallis and pairwise-Wilcox (with BH adjustment) tests, if populations were not distributed normally and/or showed heterogeneity in variance. Differences at *p* ≤ 0.05 were considered to be statistically significant.

## 3. Results and Discussion

### 3.1. Oenological and Chromatic Parameters in Wines

All wines considered in the present research presented pH of 3.5–4.0, alcohol strengths ranging between 11.6% and 14.3% vol., while titratable and volatile acidities varied from 2.5 to 3.4 g eq. H_2_SO_4_/L wine and from 0.3 to 0.7 g eq. H_2_SO_4_/L wine, respectively (Appendix A).

Regarding the chromatic parameters, color intensity ranged between a minimum value of 0.5 ± 0.0 AU for HG-D wine and a maximum of 1.7 ± 0.0 AU for both HG-E and HG-I wines. In contrast, hue values (0.6–0.8 AU) did not differ significantly among the wines considered (Appendix A).

### 3.2. Total Phenolic, Proanthocyanidin and Anthocyanin Content

Results of total phenolics, total proanthocyanidins, and total anthocyanins of hybrid monovarietal wines are depicted in Figure 1. It should be mentioned that this is the first time that wines made from those new hybrid grape varieties have been analyzed. Total phenolics, proanthocyanidins, and anthocyanins ranged, respectively, from 1547 to 3418 mg GAE/L wine, from 1.4 to 4.9 g CatE/L wine, and from 186 to 561 mg MlvE/L wine.

On the one hand, HG-B, HG-E, and HG-I wines displayed the highest total phenolic and total proanthocyanidin contents, whereas the greatest total anthocyanin values were observed for the wine elaborated with HG-H hybrid grapes (*p* < 0.05). On the other hand, both HG-D and HG-F grapes led to the poorest wines in terms of total phenolic, proanthocyanidin, and anthocyanin contents. The other hybrid grape with 99.2% of *Vitis vinifera* genome (HG-G) presented intermediate values in the upper side of the reported experimental ranges.

As observed in Figure 1, all these results were consistent with the order of magnitude of experimental values previously reported in the literature (Table 2) for monovarietal wines of international red grape varieties such as Cabernet Sauvignon (2346 ± 673 mg GAE/L wine, 1.8 ± 0.7 g CatE/L wine, 207 ± 126 mg MlvE/L wine), Merlot (2345 ± 915 mg GAE/L wine, 1.9 ± 0.9 g CatE/L wine, 171 ± 111 mg MlvE/L wine), or Syrah (2156 ± 537 mg GAE/L wine, 2.1 ± 1.0 g CatE/L wine, 207 ± 94 mg MlvE/L wine). The proposed wide ranges of total phenolic, proanthocyanidin, and anthocyanin contents of wines made from these three well-known grape varieties are a direct result of the different vintages (1978–2013), geographical origins (a total of 17 different vinegrowing countries), and viticultural/enological conditions considered in that bibliographic revision.

Apart from HG-D and HG-F, all wine samples showed phenolic, proanthocyanidin, and/or anthocyanin contents above the mean bibliographic values of international grape varieties. As observed in Figure 1, HG-B, HF-E, and HG-1 presented almost 1.5-fold times more phenolics and 2-fold times more proanthocyanidins than the mean values for Cabernet Sauvignon, Merlot, and Syrah, likewise HG-H showed ~2.5-fold times greater anthocyanin levels comparing to the three known international varieties. These results demonstrate the promising phenolic potential of these new hybrid grape varieties in red wine production.

### 3.3. Flavan-3-ol Composition of Wines

The monomeric and dimeric flavan-3-ol composition of wine samples is described in Table 3. All of the wines were analyzed to identify and quantify the monomers (+)-catechin (Cat) and (−)-epicatechin, and the dimers B1, B2, B3, and B4.

Adding up the individual concentrations of each of the above-mentioned compounds, the total content of flavan-3-ols in wines samples ranged from 29.5 to 121.6 mg Cat/L wine. Specifically, HG-H and HG-I hybrid grapes led to wines with the highest flavan-3-ol content. Meanwhile, both HG-A and HG-F wines presented the lowest values. These results fell in line with the wide ranges previously reported in the literature (Table 2) for the flavan-3-ol content of red monovarietal wines elaborated with Cabernet Sauvignon (18–255 mg/L wine) and Merlot (28–219 mg/L wine), but were slightly lower than those observed for Syrah variety (102–255 mg/L wine) [18,19,20,21,22,23,24,25,26,27].

In terms of distribution, the monomeric fraction was greater than the dimeric one for all wines, representing from 57% to 69% of the total flavan-3-ols quantified depending on the hybrid grape variety considered. The predominance of the monomeric fraction has been previously observed for monovarietal Cabernet Sauvignon, Merlot, and Syrah wines [19,20,24,28].

A general ranking order of the individual flavan-3-ols was identified throughout all the hybrid wines, apart from that elaborated with HG-A hybrid grape variety. The monomer (+)-catechin was the most abundant flavan-3-ol, representing from 42% to 50% of the total flavan-3-ol content, and the major monomer, accounting for 55–76% of the monomeric fraction. In contrast, (−)-epicatechin predominated over (+)-catechin in the case of HG-A wine, with a 2.0-fold higher concentration. Both observations are in agreement with the literature, since both behaviors have been previously described for monovarietal wines of the most international red grape varieties (Table 2).

Regarding the oligomers, the procyanidins B1 and B3 co-eluted. Thus, no data about their individual concentration may be given. The procyanidin dimer B2 displayed moderate values from 2.8 to 25.3 mg Cat/L wine for HG-F and HG-I, respectively. Meanwhile, as previously underlined by Landrault et al. [27] and Monagas et al. [24], the procyanidin B4 was present as a minor constituent, with concentrations lower than 7% of the total flavan-3-ol content.

Significant differences (*p* < 0.05) noted in terms of quantification of the individual compounds revealed a particular monomeric and oligomeric flavan-3-ol composition for wines elaborated with each hybrid grape variety.

### 3.4. Anthocyanin Composition of Wines

The anthocyanin composition of wine samples is also shown in Table 3. All of the wines were analyzed by HPLC to identify and quantify the anthocyanin 3-*O*-monoglucosides (delphinidin, Dp; cyanidin, Cy; petunidin, Pt; peonidin, Pn; and malvidin, Mlv), as well as the acetylated and *p*-coumaroylated forms of Pn and Mlv.

The total content of anthocyanins in wine samples, calculated by adding up the individual concentration of each above-mentioned compound, ranged from 68 mg/L to 306 mg/L wine for HG-I and HG-H hybrid grape varieties, respectively. These values were included within the broad range described in the literature (Table 2) for monovarietal wines of international red grape varieties such as Cabernet Sauvignon (96–699 mg/L wine) or Merlot (48–313 mg/L wine) [18,20,21,22,25,28,34,38,40]. The above-mentioned experimental interval revealed significant differences among the nine hybrid grape varieties in study (*p* ≤ 0.05). Specifically, wines from HG-B, HG-F, and HG-I were found to be particularly poor in total anthocyanin content, whereas HG-H, HG-A, and HG-C hybrid grape varieties, in this order, presented the greatest values.

In terms of distribution of the individual compounds, the simple 3-*O*-glucosides were the most abundant anthocyanins (64–87% of the total anthocyanins quantified), followed by the acetyl (7–29% of the total anthocyanins quantified), and *p*-coumaroyl (4–11% of the total anthocyanins quantified) glucosides, in that order. This behavior has been previously reported in the literature for most *V. vinifera* grapes used in winemaking [18,20,21,22,38,40].

A general anthocyanin trend persisted throughout the entire set of new hybrid grape varieties considered (Table 3), with malvidin forms exhibiting the greatest concentrations (*p* ≤ 0.05) within each anthocyanin family. As expected, Mlv-3*O*-glc was the most abundant anthocyanin, displaying values from 28.1 mg/L to 145.0 mg/L wine and accounting for 40–62% of the total anthocyanin content. Knowing that there is a direct relationship between the color of high quality wine and the content of Mlv-3*O*-glc in the berry skin, this observation is very important for the aging potential of hybrids wines. Depending on the hybrid grape variety, Pt-3*O*-glc or Mlv-3*O*-acglc were the second main component, contributing up to 15% and 27% of the total anthocyanin content, respectively. The anthocyanins Pn-3*O*-glc, Dp-3*O*-glc, and Mlv-3*O*-cmglc exhibited moderate values, whereas the compounds Cy-3*O*-glc, Pn-3*O*-acglc, and Pn-3*O*-cmglc were the minor constituents in all cases, with concentrations lower than 5% of the total anthocyanin content. These results were in broad agreement with the ranking order of the anthocyanin compounds reported for *Vitis vinifera* grape varieties in the cited bibliographic references.

Significant differences (*p* ≤ 0.05) in terms of quantification of the individual compounds revealed a particular anthocyanin profile for the different red monovarietal wines elaborated with the new hybrid grape varieties, giving a touch of uniqueness to each of them.

### 3.5. Total Antioxidant Capacity

Results of total antioxidant capacity measured by ABTS, CUPRAC, and FRAP are depicted in Figure 1D. Similar behavior patterns were observed for the three assays (Pearson’s correlation coefficient *r* ≥ 0.97, *p* < 0.05), regardless of their action mechanism. Specifically, ABTS values ranged from 17.2 ± 0.9 mmols Trolox eq./L wine to 33.3 ± 0.8 mmols Trolox eq./L wine; CUPRAC values varied between 12.0 ± 0.5 mmols Trolox eq./L wine and 30.6 ± 1.1 mmols Trolox eq./L wine; and FRAP values ranged between 9.9 ± 0.6 mmols Trolox eq./L wine and 25.7 ± 1.0 mmols Trolox eq./L wine. Both HG-D and HG-F wines showed the lowest antioxidant capacities, whereas both HG-E and HG-I presented the greatest values. All of these results presented the same order of magnitude as the total antioxidant capacity values reported in the literature for the international Cabernet Sauvignon, Merlot, and Syrah grape varieties (12.5 ± 9.7, 11.3 ± 9.2, and 15.6 ± 8.1 mmols Trolox/L wine, respectively, measured by different spectrophotometric and fluorometric assays, Table 4). Nevertheless, most of the new hybrid grape varieties considered in the present research exhibited total antioxidant capacities significantly higher than those bibliographic values (*p* ≤ 0.05).

### 3.6. Fruity Volatile Composition of Wines

Esters, enzymatically produced during yeast fermentation, significantly contribute to the typical fruity and floral character of young wines. Even if these volatile compounds are present at concentrations well below their perception thresholds, they are known to play a key role in the fruity aromatic expression of red wines, via synergistic phenomena [47]. The term ‘esters’ comprises different families of compounds, of which the ethyl esters of straight-chain fatty acids, the higher alcohol acetates, and the ethyl branched acid esters, are the most abundant, in that order. 

Their concentration in wine mainly depends on factors such as yeast strain, fermentation temperature, aeration degree, and/or sugar levels [48], as well as the quantity of the corresponding precursors originally present in the grape. For this reason, a broad bibliographic range of values is described in the literature for the different esters families and/or grape variety.

Fruity aroma profiles of the red monovarietal wines elaborated with the new hybrid grape varieties are depicted in Figure 2 (quantification of fruity volatiles by ester family). The HG-I wine showed the greatest total fruity volatile content (1681 ± 10 µg/L) (*p* ≤ 0.05), nearly followed by HG-B wine (1432 ± 4 μg/L). Meanwhile, the hybrid grape varieties HG-D, HG-G, and HG-H led to wines with the lowest values (939–970 μg/L). As observed in Figure 2, the HG-B, HG-E, and HG-I presented, respectively, the greatest contents of ethyl esters of straight-chain fatty acids (1.1–1.8-fold times higher), alcohol acetates (1.1–1.7-fold times higher) and ethyl branched acid esters (2.2–6.1-fold times higher).

A general trend persisted throughout all of the hybrid grape varieties, except for HG-I wine, with ethyl esters of straight-chain fatty acids (530–929 μg/L) standing out clearly as the main components, followed by higher alcohol acetates (226–391 μg/L) and ethyl branched acid esters (82–500 μg/L). All of these experimental results showed the same order of magnitude as those previously reported in the literature for volatile ester composition of Cabernet Sauvignon, Merlot, and/or Syrah monovarietal wines of different provenance (Australia, Brazil, France, Italy, Switzerland, United States) (Appendix A) [11,49,50,51,52].

A detailed quantification of individual fruity volatiles is also available in the Appendix A. All hybrid grape varieties presented ethyl propanoate, reported to provide the wine with ripe strawberry-like aromas, as the most abundant ethyl ester of straight-chain fatty acids. According to the literature, ethyl hexanoate, ethyl octanoate, or ethyl decanoate have always been found as the main components of this ester family [11,50,51]. It is noteworthy to mention, in the case of hybrid wines, a different ranking order of the second main ethyl esters of straight-chain fatty acids was observed according to the grape crossings performed (Table 1). In the case of HG-A wine, ethyl octanoate (hints of ripe fruit) was the second main component of that ester family. For both HG-B and HG-C, as well as for all four wines elaborated with hybrid grapes with 99.2% of *Vitis vinifera* genome (HG-F, HG-G, HG-H, and HG-I), ethyl hexanoate (pineapple, green apple, and strawberry aromas) predominated over ethyl octanoate. Meanwhile, a similar content of both esters was observed for HG-D and HG-E.

With regard to the higher alcohol acetates and the ethyl branched acid esters, isoamyl acetate, characterized by banana notes, and ethyl isobutyrate, described by strawberry, kiwi, and lemon odors, were, respectively, the major volatiles in all cases. Their predominance within the corresponding ester families has been previously underlined in the literature [11,50,51] for Cabernet Sauvignon, Merlot, and/or Syrah monovarietal wines.

### 3.7. Sensory Analysis

According to sensory analysis, significant differences (*p* < 0.05) were observed for global balance, astringency, and roundness (Appendix A). For each one of these descriptors, only certain wines were completely differentiated (the rest did not show significant differences). Specifically, monovarietal wine from HG-B hybrid grapes was perceived as the most astringent one, and characterized by the lowest roundness and global balance among the tasted wines. Both HG-A and HG-D grapes led to wines with the lowest astringency. Tasters well appreciated the greatest roundness and global balance of monovarietal wine from HG-E hybrid grapes.

No significant differences were observed with regard to the fruitiness of wines (*p* > 0.05). For further elucidation of the aromatic complexity and sensory properties of these monovarietal red wines produced from new hybrid grape varieties, next step might be to compare their organoleptic characteristics to those of *Vitis vinifera* red wines that consumers are used to, by applying the Pivot profile method and Polarized Sensory Positioning test.

## 4. Conclusions

This is the first time in the literature that wines made from these new hybrids grape varieties, so-called Bouquet varieties, have been analyzed. In a first approach, HG-I (99.2% of *Vitis vinifera* genome, Cabernet Sauvignon in last backcross) exhibited the highest phenolic and aromatic values among the disease resistant grape varieties considered.

Although further studies are required to obtain a more complete characterization of the organoleptic profile of these wines compared to that of *Vitis vinifera* red wines that consumers are used to, the present study highlights some specific sensory characteristics regarding global balance, astringency, and roundness.

Results showed that these new red varieties may have enough potential to produce quality wines, as their phenolic and volatile composition is close to that of the commonly used monovarietal red wines. For this reason, as well as for their resistance to cryptogamic diseases, the present research encourages the wine industry to host these new hybrid grapes to ensure not only the quality but also the quantity of future wines. In this sense, their use in winemaking might provide quality wines that diversify the wine offer in an increasingly global and homogeneous oenological market. Further studies are needed in order to approach their aging potential.

## Figures and Tables

**Figure 1 biomolecules-09-00793-f001:**
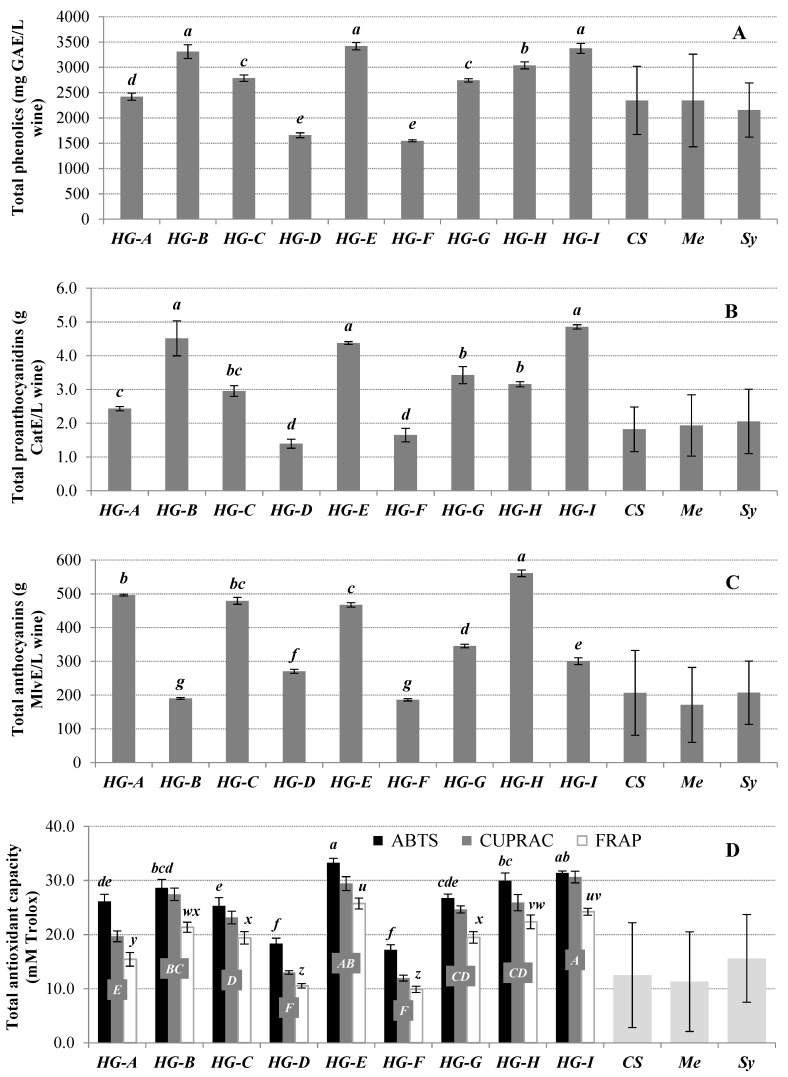
Total phenolics (**A**), total proanthocyanidins (**B**), total anthocyanins (**C**), and total antioxidant capacity (**D**) of monovarietal red wines from the new hybrid grape (HG) varieties compared to the corresponding bibliographic ranges found for monovarietal wines made from international red grape varieties (CS, Cabernet Sauvignon; Me, Merlot; Sy, Syrah). For (**A**–**C**), lower case letters *a*–*g* show significant differences among hybrid grape varieties (*p* < 0.05). For (**D**), lower case letters *a*–*f*, capital letters *A*–*F* and lower case letters *u*–*z* show significant differences among hybrid grape varieties (*p* < 0.05) for ABTS, CUPRAC, and FRAP results, respectively.

**Figure 2 biomolecules-09-00793-f002:**
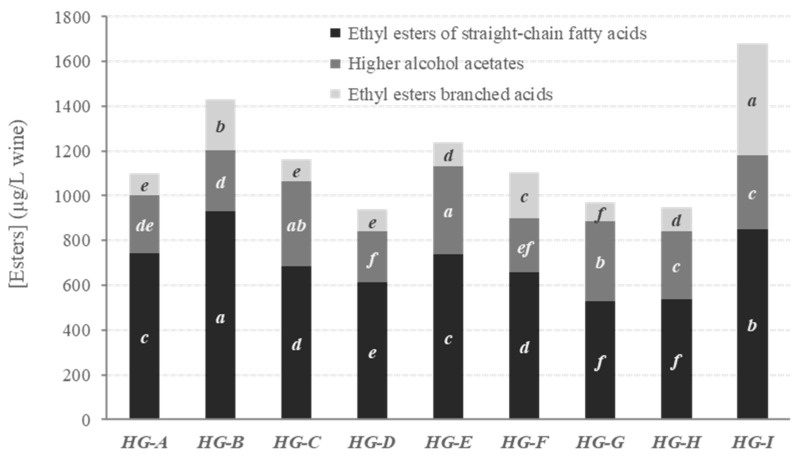
Fruity volatile profile of monovarietal red wines from the new hybrid grape (HG) varieties. Lower case letters *a*–*f* show significant differences among hybrid grape varieties for each family of esters (*p* < 0.05).

**Table 1 biomolecules-09-00793-t001:** Disease resistant Bouquet varieties considered in the present research.

Identification in the Present Research	INRA Identification ^a^	Backcross Number	Last Backcross	Estimated % of *Vitis vinifera* Genome
HG-A	3184-1-9N (G14)	BC5	A. Lavallée x 3099-10-57	98.7%
HG-B	3176-21-11N	BC5	Grenache x 3084-2-56	98.7%
HG-C	3328-306N	BC5	3082-1-49 x Marselan	98.7%
HG-D	3160-11-3N	BC5	Fer Servadou x 3090-4-25	98.7%
HG-E	3160-27-4N	BC5	Fer Servadou x 3090-4-25	98.7%
HG-F	3322-339N	BC6	3176-21-11N x Cabernet Sauvignon	99.2%
HG-G	3322-343N	BC6	3176-21-11N x Cabernet Sauvignon	99.2%
HG-H	3322-178N	BC6	3176-21-11N x Cabernet Sauvignon	99.2%
HG-I	3322-226N	BC6	3176-21-11N x Cabernet Sauvignon	99.2%

^a^ Adapted from [7]. HG, hybrid grape.

**Table 2 biomolecules-09-00793-t002:** Bibliographic data about total phenolics, total proanthocyanidins (both spectrophotometric and quantified by HPLC), and total anthocyanins (both spectrophotometric and quantified by HPLC) for monovarietal wines made from international red grape varieties (Cabernet Sauvignon, Merlot, Syrah).

Bibliographic Reference	Wine Characteristics				Total Phenolics ^a^	Total Proantho-Cyanidins ^b^	Total Anthocyanins ^c^	Total Flavan-3-ols Quantified by HPLC ^c^	Total Anthocyanins Quantified by HPLC ^c^
	Geographical Origin	Vintage	Grape Variety						
[18]	Maipo Valley (Chile)	2010	CS		894 ± 4	1.7 ± 0.2	486 ± 2	116 ± 9	487
			Merlot		795 ± 4	1.8 ± 0.2	393 ± 4	97 ± 4	313
[19]	Bordeaux (France)	1978–2005	CS		1579–3188	1.2–2.2		18–97	
		1979–2003	Merlot		1244–2544	1.2–2.1		28–91	
[20]	Mendoza (Argentina)	2010	CS		3378 ± 370	3.9 ± 0.4	682 ± 101	192 ± 10	327
			Merlot		3448 ± 372	4.4 ± 0.5	645 ± 38	190 ± 13	273
			Syrah		1586 ± 51	1.9 ± 0.2	301 ± 19	110 ± 17	168
[21]	China	2007	CS					97–246	253–467
[22]	San Juan (Argentina)	2014	CS					169 ± 3	101
			Merlot					140 ± 4	73
			Syrah					102 ± 3	161
[23]	China	2011	CS		2631 ± 42	1.0 ± 0.1		190	
			Merlot		2076 ± 7	1.0 ± 0.0		185	
[24]	Navarra (Spain)	2000	CS					90 ± 2	
[25]	Montenegro	2015	CS				353 ± 77	35–92	231–489
[26]	China	2010	CS		1130–2710		262–400	30–255	
			Merlot		860–1656		158–350	42–91	
[27]	France	1993–1999	CS		1842–2532			151–225	
		1993–1999	Merlot		1783–2698			115–219	
		1998–1999	Syrah		2200–2590			149–255	
[28]	Greece	2002	CS		2481 ± 10				699
			Syrah		1920 ± 19				458
[29]	Navarra (Spain)	2003	CS		3610				
			Merlot		2920				
[30]	Sicily (Italy)	2002–2004	CS		2380–3580				
			Merlot		2999–3360				
			Syrah		3000–3410				
[31]	Australia	2003–2005	CS		2382 ± 490	1.5 ± 0.4	190 ± 54		
		2003–2005	Merlot		2518 ± 506	1.3 ± 0.3	134 ± 38		
		2002–2005	Syrah		2064 ± 258	1.3 ± 0.2	198 ± 93		
[32]	Italy (and others)	2009	Merlot		2791 ± 1711				
			Syrah		1991 ± 234				
[33]	Uruguay	2001–2002	CS			1.7–2.4	349–563		
			Merlot			1.5–2.0	227–402		
[34]	Uruguay	2001–2002	CS						181–230
			Merlot						279–296
[35]	Romania	2011–2013	CS		1986–2758		259–479		
[36]	Brazil	2002–2007	CS		1260–1894				
		2005–2007	Merlot		1318–1844				
		2005–2007	Syrah		1753–1914				
[37]	Brazil, Argentina, Chile	2005–2007	CS						
		2002–2007	Merlot						
		2006–2007	Syrah						
[38]	La Mancha (Spain)	not specified	CS						206
		not specified	Syrah						358
[39]	Romania	2006–2008	CS		1896–4263	1.0–2.3	84–216		
			Merlot		1913–3863	1.2–2.4	63–281		
[40]	Macedonia	2006–2008		CS					96–351
				Merlot					48–194
[41]	Australia, Chile, France, Spain, USA	2003–2005		CS	1453–2912				
	France, Germany, Italy, Spain	2004–2005		Merlot	1447–2100				
[42]	Australia	2005–2007		CS		1.8–2.8			
				Syrah		1.3–2.9			
[43]	Serbia	2012		CS	1100				
				Merlot	890				
				Syrah	670				
[44]	Croatia	2002		CS	1400				
				Merlot	1300				
[45]	Ontario (Canada)	2002		CS	2005				
[46]	Thessaloniki (Greece)	2004		CS		2.8–4.4			
				Merlot		1.7–5.1			
				Syrah		1.7–4.7			

CS, Cabernet Sauvignon; C, (+)-catechin; EC, (−)-epicatechin. ^a^ Total phenolics expressed in mg gallic acid equivalents/L wine. ^b^ Total proanthocyanidins expressed in g/L wine. ^c^ Expressed in mg/L wine.

**Table 3 biomolecules-09-00793-t003:** Flavan-3-ol and anthocyanin profiles of monovarietal red wines from the new hybrid grape (HG) varieties. For each individual compound, lower case letters *a*–*h* show significant differences among hybrid grape varieties (*p* < 0.05).

	HG-A			HG-B			HG-C			HG-D			HG-E			HG-F			HG-G			HG-H			HG-I		
Flavan-3-ols ^a^																											
(+)-catechin	8.4	±	0.3 *h*	30.1	±	0.6 *f*	37.6	±	0.3 *d*	38.2	±	0.2 *cd*	31.0	±	0.2 *e*	14.4	±	0.1 *g*	43.8	±	0.2 *b*	38.9	±	0.1 *c*	46.1	±	0.3 *a*
(−)-epicatechin	17.1	±	0.4 *c*	9.7	±	0.3 *e*	13.7	±	0.1 *d*	16.7	±	0.1 *c*	14.1	±	0.1 *d*	4.6	±	0.2 *f*	13.9	±	0.1 *d*	18.3	±	0.1 *b*	37.6	±	0.5 *a*
procyanidin dimers B1+B3	7.4	±	0.1 *g*	17.1	±	0.3 *b*	18.3	±	0.1 *a*	13.5	±	0.1 *d*	12.8	±	0.1 *e*	6.1	±	0.1 *h*	16.5	±	0.1 *c*	18.6	±	0.0 *a*	10.0	±	0.2 *f*
procyanidin dimer B2	5.3	±	0.2 *f*	9.2	±	0.3 *e*	13.1	±	0.2 *b*	10.0	±	0.1 *d*	10.2	±	0.2 *d*	2.8	±	0.1 *g*	8.8	±	0.1 *e*	12.1	±	0.0 *c*	25.3	±	0.3 *a*
procyanidin dimer B4	2.7	±	0.1 *d*	4.0	±	0.1 *b*	5.7	±	0.2 *a*	3.5	±	0.1 *c*	3.2	±	0.0 *c*	1.6	±	0.0 *e*	4.3	±	0.1 *b*	4.4	±	0.0 *b*	2.6	±	0.3 *d*
Total flavan-3-ols ^b^	40.9	±	0.6	70.2	±	0.8	88.3	±	0.4	81.9	±	0.2	71.2	±	0.3	29.5	±	0.2	87.3	±	0.3	92.3	±	0.2	121.6	±	0.7
Anthocyanins ^c^																											
Dp-3*O*-glc	30.3	±	0.0 *a*	6.1	±	0.1 *f*	9.1	±	0.0 *c*	6.3	±	0.0 *f*	8.1	±	0.1 *d*	7.3	±	0.1 *e*	7.5	±	0.0 *e*	28.3	±	0.2 *b*	3.5	±	0.1 *g*
Cy-3*O*-glc	5.1	±	0.0 *a*	3.7	±	0.0 *c*	3.7	±	0.0 c	3.6	±	0.0 *cd*	3.7	±	0.0 *c*	3.7	±	0.1 *c*	3.5	±	0.0 *d*	4.3	±	0.0 *b*	0.6	±	0.1 *e*
Pt-3*O*-glc	37.0	±	0.1 *a*	6.9	±	0.1 *f*	12.8	±	0.2 *c*	8.7	±	0.1 *e*	10.6	±	0.0 *d*	8.6	±	0.2 *e*	11.0	±	0.1 *d*	33.2	±	0.2 *b*	6.2	±	0.4 *g*
Pn-3*O*-glc	13.9	±	0.3 *a*	7.3	±	0.1 *d*	6.5	±	0.0 *ef*	9.7	±	0.1 *c*	10.1	±	0.2 *c*	6.0	±	0.0 *f*	6.8	±	0.2 *de*	12.7	±	0.3 *b*	1.9	±	0.1 *g*
Mlv-3*O*-glc	125.4	±	1.0 *b*	28.1	±	0.2 *h*	116.8	±	0.7 *c*	98.1	±	0.6 *d*	99.8	±	0.2 *d*	45.9	±	0.2 *f*	92.4	±	1.6 *e*	145.0	±	2.1 *a*	38.6	±	0.8 *g*
Pn-3*O*-acglc	4.0	±	0.0 *d*	3.7	±	0.1 *de*	4.7	±	0.0 *b*	4.3	±	0.0 *c*	4.4	±	0.0 c	3.5	±	0.0 *e*	4.9	±	0.0 *b*	5.9	±	0.2 *a*	0.6	±	0.0 *f*
Mlv-3*O*-acglc	12.0	±	0.1 *f*	7.4	±	0.2 g	64.1	±	0.0 *a*	14.1	±	0.0 *e*	22.4	±	0.0 *d*	7.1	±	0.1 *g*	40.3	±	0.2 *c*	56.1	±	0.2 *b*	14.0	±	0.0 *e*
Pn-3*O*-cmglc	4.4	±	0.0 *b*	3.6	±	0.0 *e*	3.8	±	0.0 *d*	4.0	±	0.0 *d*	4.1	±	0.1 *c*	3.4	±	0.0 *f*	4.3	±	0.0 *b*	4.9	±	0.0 *a*	0.3	±	0.0 *g*
Mlv-3*O*-cmglc	11.1	±	0.1 *c*	4.1	±	0.0 *f*	12.3	±	0.2 *b*	9.6	±	0.1 *d*	11.0	±	0.3 *c*	4.6	±	0.1 *e*	16.4	±	0.0 *a*	15.8	±	0.6 *a*	2.6	±	0.0 *g*
Total anthocyanins ^d^	243.2	±	1.0	70.8	±	0.3	233.7	±	0.7	158.5	±	0.6	174.3	±	0.4	90.1	±	0.4	187.2	±	1.6	306.1	±	2.3	68.3	±	0.9

^a^ Expressed in mg catechin/L wine. ^b^ Total flavan-3-ols calculated as the sum of (+)-catechin, (−)-epicatechin, B1, B2, B3, and B4 individual contents. ^c^ Expressed in mg malvidin/L wine. ^d^ Total anthocyanins calculated as the sum of all anthocyanin individual contents. HG, hybrid grape; glc, monoglucoside; acglc, 6″-acetylglucoside; cmglc, 6″-*p*-coumaroylglucoside; Dp, delphinidin; Cy, cyanidin; Pt, petunidin; Pn, peonidin; Mlv, malvidin. Letters following the values in each row show the significant differences among hybrid grape varieties (*p* < 0.05).

**Table 4 biomolecules-09-00793-t004:** Bibliographic data about antioxidant capacity for monovarietal wines made from international red grape varieties (Cabernet Sauvignon, Merlot, Syrah).

Bibliographic Reference	Wine Characteristics			Methodology	Total Antioxidant Capacity ^a^		
	Geographical Origin	Vintage	Grape Variety				
[22]	San Juan (Argentina)	2014	CS	FRAP	8.2	±	0.4
				ABTS	14.1	±	1.0
				DPPH	11.9	±	1.0
			Merlot	FRAP	9.0	±	0.1
				ABTS	18.5	±	0.5
				DPPH	11.9	±	0.8
			Syrah	FRAP	8.5	±	0.2
				ABTS	17.3	±	0.3
				DPPH	12.8	±	1.6
[25]	Montenegro	2015	CS	ABTS	16.3	±	5.2
[26]	China	2010	CS	DPPH	4.6	−	6.2
				CUPRAC	10.0	−	20.0
			Merlot	DPPH	3.9	−	5.3
				CUPRAC	9.0	−	17.5
[27]	France	1993–1999	CS	ABTS	16.5	−	29.9
		1993–1999	Merlot	ABTS	15.3	−	22.2
		1998–1999	Syrah	ABTS	19.7	−	22.1
[30]	Sicily (Italy)	2002–2004	CS	no specified	1.4	−	5.6
			Merlot	no specified	2.2	−	4.9
			Syrah	no specified	1.2	−	5.8
[31]	Australia	2003–2005	CS	DPPH	15.9	±	2.3
				ABTS	18.9	±	3.0
		2003–2005	Merlot	DPPH	15.2	±	3.1
				ABTS	17.7	±	4.8
		2002–2005	Syrah	DPPH	13.0	±	2.2
				ABTS	16.9	±	5.1
[32]	Italy (and others)	2009	Merlot	ABTS	17.5	±	8.9
			Syrah	ABTS	13.3	±	3.0
[36]	Brazil	2002–2007	CS	ORAC	20.7	−	35.7
		2005–2007	Merlot	ORAC	16.3	−	35.4
		2005–2007	Syrah	ORAC	28.0	−	38.6
[37]	Brazil. Argentina, Chile	2005–2007	CS	ORAC	28.8	−	33.4
		2002–2007	Merlot	ORAC	26.0	−	33.7
		2006–2007	Syrah	ORAC	29.0	−	31.5
[39]	Romania	2006–2008	CS	ABTS	1.1	−	1.3
			Merlot	ABTS	1.0	−	1.3
[40]	Macedonia	2006–2008	CS	DPPH	10.3	−	11.2
			Merlot	DPPH	12.3	−	13.3
[41]	Different countries	2003–2005	CS	ABTS	7.7	−	16.6
				FRAP	7.0	−	15.2
		2004–2005	Merlot	ABTS	7.5	−	11.2
				FRAP	6.9	−	9.7
[43]	Serbia	2012	CS	DPPH	8.0		
			Merlot	DPPH	6.5		
			Syrah	DPPH	4.3		

^a^ Total antioxidant capacity expressed in mmols Trolox equivalents/L wine.

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
