# Peer review of "Disease Resistant Bouquet Vine Varieties: Assessment of the Phenolic, Aromatic, and Sensory Potential of Their Wines"

_biomolecules, 2019, doi:10.3390/biom9120793_

Round 1

Reviewer 1 Report

The work s written correctly, legibly, well edited.

I assess the wide discussion of the results obtained positively.

The results obtained are valuable mainly from the application point of view.

In my evaluation the only week point is the description of the vinification process in the methods part of the manuscript. This details are important and definitely can also influence the evaluated oenological parameters.

Lak of following information:

What volume of fermenters was applied?

How many times (and when) racking of the young wine was performed?

Were the malolactic bacteria preparates used? Or it was spontaneous process?

How long did the wines matured at 16^C before the final assessment?

In my opinion, after this corrections/supplements the manuscript is worth for publishing.

Author Response

Response to reviewer [1]

[1] In my evaluation, the only weak point is the description of the vinification process in the methods part of the manuscript. These details are important and definitely can also influence the evaluated oenological parameters. Lack of following information: What volume of fermenters was applied? How many times (and when) racking of the young wine was performed? Were the malolactic bacteria preparates used? Or it was spontaneous process? How long did the wines matured at 16 °C before the final assessment?

Following the suggestion of the reviewer, some additional information about the vinification process has been included in the sub-section 2.2 Red wine vinification. Fermenters had a volume of 50L. Wines were inoculated with lactic acid bacteria to perform malolactic fermentation, and racked just once after malolactic fermentation. All wines were stored at 16 °C for two months before their phenolic, aromatic and sensory analysis.

Thus, the sub-section 2.2 has been modified as follows ‘Microvinification assays were carried out separately for each hybrid grape to obtain the corresponding monovarietal wines. Hybrid grapes were manually harvested at maturity during the 2016 vintage. Grapes were crushed and destemmed the day of harvest. Potassium metabisulphite (3 g/hL) was added during the transfer of must to stainless steel tank (50 L) and Saccharomyces cerevisiae (Anchor NT 202, La Littorale) was included to perform alcoholic fermentation at 23 – 25 °C.

Wine was inoculated with lactic acid bacteria (Maloferm Fruity, Oenobrands) to perform malolactic fermentation (MLF). It was conducted in all cases at a constant temperature of 22 °C and extended for a variable period of time depending on the hybrid grape considered (from seven to 35 days).

Once the MLF concluded (malic acid content ≤ 0.2 g/L), all wines were immediately racked, stabilized and sulfitated prior to bottling and storage at 16 °C until further analysis.’.

Reviewer 2 Report

The article “Disease resistant Bouquet vines varieties: assessment of the phenolic, aromatic and sensory potential of their wines” by González-Centeno and colleagues analyses the physico-chemical and organoleptic characteristics of 9 wines obtained from hybrid grapes and compare them with similar red wines from the literature. Despite some minor English and editing errors (e.g. lines 20, 51, etc), the article is quite well-written and organized. My major concern is that maybe a journal such as Foods would have been more appropriate, but that is an editorial choice. Other considerations are:

It is not clear to me why some analyses were performed in duplicate (i.e. sections 2.5, 2.6, 2.8) or sometimes it is not specified at all the number of replicates (i.e. section 2.4). They should have run at least 3 replicates for each analysis.

Also, why the author did not try to perform a multivariate principal component analysis combining all the data (at least 3 samples for each type)? It could have evidenced better similarities and differences among the 9 wines overall.

Other suggestions are:

Line 60: I would add the following reference to strengthen what stated: Foods 2017, 6, 24; doi:10.3390/foods6040024

Lines 73-79 report historical info and they should be moved to the Results and Discussion section.

Line 102: the correct English acronym is “FT-IR”.

Author Response

Response to reviewer [2]

[1] It is not clear to me why some analyses were performed in duplicate (i.e. sections 2.5, 2.6, 2.8) or sometimes it is not specified at all the number of replicates (i.e. section 2.4). They should have run at least 3 replicates for each analysis.

We apologize for this misunderstanding. It is just a spelling mistake: all analyses were performed at least in triplicate. Only results of both total phenolic content and total antioxidant capacity, which are spectrophotometrically evaluated with an automated microplate reader, are expressed as a mean of six determinations. The manuscript has been corrected accordingly.

[2] Why the author did not try to perform a multivariate principal component analysis combining all the data (at least 3 samples for each type)? It could have evidenced better similarities and differences among the 9 wines overall.

We agree with Reviewer#2 that principal component analysis may give a global overview of the similarities and differences among the 9 monovarietal wines from hybrid grapes. Nevertheless, it was not the goal of this paper, which mainly wanted to compare them with monovarietal wines produced with well-known red grape varieties (Cabernet Sauvignon, Merlot, Syrah) in order to reveal their similar phenolic and aromatic potential to lead to quality wines.

In fact, a second manuscript about disease resistant grape varieties is now in process, combining a larger number of wines and different vinification techniques. In this case, results from principal component analysis will be included.

[3] Line 60: I would add the following reference to strengthen what stated: Foods 2017, 6, 24; doi:10.3390/foods6040024.

As requested, the reference has been included to strengthen what stated.

[4] Lines 73-79 report historical info and they should be moved to the Results and Discussion section.

The authors agree with Reviewer#2 that lines 73-79 are not well placed in the Materials and methods section. Nevertheless, the authors think that it is more appropriate to include this historical information in the Introduction section to complete lines 52-54.

[5] Line 102: the correct English acronym is “FT-IR”.

The acronym of Infrared Spectrometry with Fourier Transformation has been corrected.